# SerpinB3 Upregulates Low-Density Lipoprotein Receptor-Related Protein (LRP) Family Members, Leading to Wnt Signaling Activation and Increased Cell Survival and Invasiveness

**DOI:** 10.3390/biology12060771

**Published:** 2023-05-26

**Authors:** Santina Quarta, Andrea Cappon, Cristian Turato, Mariagrazia Ruvoletto, Stefania Cannito, Gianmarco Villano, Alessandra Biasiolo, Maristella Maggi, Francesca Protopapa, Loris Bertazza, Silvano Fasolato, Maurizio Parola, Patrizia Pontisso

**Affiliations:** 1Department of Medicine, University of Padova, 35128 Padua, Italy; 2Department of Molecular Medicine, University of Pavia, 27100 Pavia, Italy; 3Department of Clinical and Biological Sciences, University of Torino, 10124 Turin, Italy; 4Department of Surgical, Oncological and Gastroenterological Sciences, University of Padova, 35128 Padua, Italy

**Keywords:** β-catenin signaling, cell invasiveness, Wnt cell membrane co-receptors

## Abstract

**Simple Summary:**

The Wnt-β-catenin signaling regulates liver homeostasis and repair in adulthood, while its abnormal regulation is involved in the development of several chronic diseases and tumors. SerpinB3 has been shown to induce β-catenin, and both molecules are overexpressed in tumors, particularly in those with poor prognoses. The aim of our study was to evaluate the ability of SerpinB3 to modulate the Wnt pathway in liver cancer and monocytic cells, the main type of inflammatory cells in the tumor microenvironment. We have demonstrated that SerpinB3 modulates the Wnt cascade upregulating the Wnt co-receptors, low-density lipoprotein receptor-related protein 5 (LRP-5) and LRP-6, as well as LRP-1, implicated in cell survival and invasiveness. These data were confirmed in experimental carcinogenesis. In conclusion, the upregulation of LRP family members by SerpinB3 determines Wnt signaling activation and increased cell survival and invasiveness.

**Abstract:**

Abnormal activation of the Wnt-β-catenin signaling cascade is involved in tumor growth and dissemination. SerpinB3 has been shown to induce β-catenin, and both molecules are overexpressed in tumors, particularly in those with poor prognoses. The aim of this study was to evaluate the ability of SerpinB3 to modulate the Wnt pathway in liver cancer and in monocytic cells, the main type of inflammatory cells in the tumor microenvironment. The Wnt cascade, Wnt co-receptors, and low-density lipoprotein receptor-related protein (LRP) members were analyzed in different cell lines and human monocytes in the presence or absence of SerpinB3. The Wnt-β-catenin axis was also evaluated in liver tumors induced in mice with different extents of SeprinB3 expression. In monocytic cells, SerpinB3 induced a significant upregulation of Wnt-1/7, nuclear β-catenin, and c-Myc, which are associated with increased cell lifespan and proliferation. In liver tumors in mice, the expression of β-catenin was significantly correlated with the presence of SerpinB3. In hepatoma cells, Wnt co-receptors LRP-5/6 and LRP-1, implicated in cell survival and invasiveness, were upregulated by SerpinB3. The LRP pan-inhibitor RAP not only induced a decrease in LRP expression, but also a dose–dependent reduction in SerpinB3-induced invasiveness. In conclusion, SerpinB3 determines the activation of the Wnt canonical pathway and cell invasiveness through the upregulation of LRP family members.

## 1. Introduction

The Wnt-β-catenin signaling pathway is a highly conserved and tightly controlled signaling pathway that regulates not only hepatobiliary development and cell fate during embryogenesis, but also liver homeostasis and repair in adulthood [1]. During chronic liver injury, when massive extracellular matrix deposition parallels the impairment of hepatocytes and biliary cell proliferation, the stem cell niche, composed of quiescent progenitor cells, undergoes activation. Hepatic progenitor cells (HPCs) can differentiate into either hepatocytes or cholangiocytes, and the Notch-Wnt axis has a crucial role in HPC activation and fate [2]. In particular, in a mouse model of chronic biliary injury, it has been demonstrated that activated stellate cells and myofibroblast secrete Notch ligands in the hepatic progenitor niche milieu are used to promote biliary regeneration. On the other hand, during chronic hepatocellular injury, in the staminal niche, there is an increased recruitment of macrophages that secrete Wnt molecules in response to dead hepatocytes phagocytosis or other stimuli, such as lipid accumulation [3], with Wnt signaling driving HPCs toward differentiation into hepatocyte phenotypes. The core of the Wnt signaling cascade is β-catenin which, in addition to its role in embryogenesis, tissue homeostasis, and cell renewal, is also involved in tumor growth and dissemination [3,4]. In liver tumors, this molecule has been shown to be overexpressed in the presence of the serine protease inhibitor SerpinB3 [5], especially in tumors with poor prognosis [6,7]. This serpin is not detectable in normal adult hepatocytes; however, chronically damaged hepatocytes express SerpinB3, and the highest levels are achieved in the most aggressive forms of HCC, with a high risk of early recurrence [7]. It has been recently described that the subset of cholangiocarcinomas with stem-like features, associated with poor prognosis, expresses high levels of this serpin [8]. In vitro studies have shown that SerpinB3 protects neoplastic cells from apoptotic cell death through its interaction with Complex 1 of the mitochondrial respiratory chain [9] and also through the inhibition of lysoptosis, conferring resistance to chemoradiation in cervical cancer [10]. On the other hand, SerpinB3 has been reported to increase cell proliferation through the induction of not only β-catenin but also of Myc oncogene, a downstream gene of the Wnt pathway [11]. In line with these findings, it is not surprising that tumors overexpressing SerpinB3 and β-catenin are those characterized by poor prognosis not only in the liver but also in other types of cancer, including colon [5] and cervical cancer [10].

Low-density lipoprotein receptors (LRPs), especially LRP-5 and LRP-6, are fundamental co-receptors for the activation of canonical Wnt signaling. Their phosphorylation recruits axin to the cytoplasmic tail of LRP-6, preventing β-catenin phosphorylation and subsequent proteasomal degradation. As a consequence, β-catenin accumulates in the cytoplasm and translocates to the nucleus, where it regulates the expression of target genes [12]. It is interesting to note that another member of the LRP family, the low-density lipoprotein receptor-related protein-1 (LRP-1) is a ubiquitous membrane receptor with scavenger and regulatory functions [13], which can bind several proteins, including extracellular matrix proteins, growth factors, proteins involved in lipoprotein metabolism, and particularly, proteases alone or complexed to protease inhibitors, including alpha-1 antitrypsin (AAT), antithrombin III (ATIII), plasminogen activator inhibitor-1 (PAI-1), and nexin-1 (PN-1) [13]. The binding of the receptor to the serpin-enzyme complex (SEC) induces pro-survival and anti-inflammatory responses through the phosphorylation of protein kinase Akt [14].

On the basis of these considerations, we have investigated whether SerpinB3 may interact with LRP family members, modulating their cellular expression and functions in liver cancer.

## 2. Materials and Methods

### 2.1. Cell Cultures

#### 2.1.1. Primary Human Monocytes

Human peripheral blood mononuclear cells (PBMCs) from healthy donors were isolated by density gradient centrifugation on a Ficoll–Paque (Merck KGaA, Darmstadt, Germany) solution at 500 rcf for 30 min. Mononuclear cells were harvested, resuspended in a medium containing 10% FCS (Merck KGaA, Darmstadt, Germany), and separated from contaminating lymphocytes by adherence to plastic (1 h at 37 °C). Adherent monocytes were extensively washed with a medium to remove residual nonadherent cells. The percentage of CD14+ cells was greater than 98%. Primary human monocytes were routinely maintained in an RPMI 1640 medium supplemented with 10% *v/v* FCS, 2 mmol/L L-glutamine, and 1% *v/v* Penicillin/Streptomycin (Merck KGaA, Darmstadt, Germany) in 5% CO_2_ in a humidified incubator at 37 °C.

#### 2.1.2. THP-1 Cell Line

The monocytic THP-1 cell line (kindly provided by Prof. F. Marra, Florence University, Florence, Italy) was routinely maintained in an RPMI 1640 medium supplemented with 10% *v/v* FCS, 2 mmol/L L-glutamine, and 1% *v/v* Penicillin/Streptomycin (Merck KGaA, Darmstadt, Germany) in 5% CO_2_ in air in a humidified incubator at 37 °C.

#### 2.1.3. HepG2 Cell Line

Hepatoma cells (HepG2 cell line) (ATCC, Manassas, VA, USA), authenticated by BMR Genomics S.4.l. (Padova, Italy), were engineered to overexpress SerpinB3 (HepG2/SB3) by transfection with a plasmid expression vector containing the human SerpinB3 gene (pCDNA3/SB3), or with the plasmid vector alone (pcDNA3.1D/V5-His-TOPOTM) as a control (HepG2/control), using Lipofectamine Reagent Plus as the transfecting agent, according to the manufacturer’s recommended indications (Invitrogen, Carlsbad, CA, USA), as previously reported [5]. Cells were maintained at 37 °C in a humidified chamber with 5% CO_2_ and cultured in a minimum essential medium with the addition of G418 as a selective agent.

### 2.2. Cell Proliferation Assays

A real-time analysis of cell proliferation was performed on the xCELLigence DP instrument (ACEA BioSciences, St. Diego, CA, USA), as previously described [15]. The E-plate was engaged into xCELLigence and background measurements of the wells were recorded before adding 5 × 10^3^ cells/well of primary human monocytes or THP1 cells supplemented with 100 ng/mL of recombinant SerpinB3 or with a medium alone as a control. Recombinant LPS-free SerpinB3 was obtained in our laboratory, as previously described [16].

Cells were allowed to settle for 30 min at room temperature before the E-plates were re-engaged onto the xCELLigence analyzer and incubated at 37 °C and 5% CO_2_. The electrical impedance values were recorded every 15 min until 5 days and used for data analysis. The results were expressed as the cell index, a unitless readout that reflects the number of cells attached at the bottom of the cell culture plate wells via gold electrodes.

An MTT assay was also performed on monocytic cells. A total of 5 × 10^3^ cells/well of primary human monocytes were seeded on 96 wells and treated with or without SerpinB3 in the presence or absence of the Wnt ICG-001 inhibitor (5 uM) for up to 8 days. Cell proliferation was measured by adding MTT (Merck KGaA, Darmstadt, Germany), and the absorbance of the resulting purple solution was spectrophotometrically measured at 570 nm.

### 2.3. Cell Invasion Assay

A real-time analysis of cell invasion was performed on the xCELLigence DP instrument (ACEA BioSciences, St. Diego, CA, USA), as previously described [17]. The surface of the upper chamber wells of a two-chamber device (CIM-plate 16) was coated with a monolayer of 1× collagen I solution (Merck KGaA, Darmstadt, Germany) in order to create a matrix suitable for the evaluation of invasive cellular activity. A medium with 10% serum was placed in the lower chamber as a chemoattractant. The two chambers are separated by a porous membrane and cells migrate through a solid matrix at the membrane where the electrodes reside. HepG2 cells were seeded (5 × 10^4^ cells/well) in a serum-free medium in the upper chamber, according to the manufacturer’s instructions (ACEA, St. Diego, CA, USA). The following treatments were carried out: (a) pre-incubation with a mouse monoclonal antibody to LRP-1 515 kDa (Meridian Life Science, Inc., Memphis, USA) (0.5 and 5 g/mL) for 1 h, or with the same concentrations of a generic mouse antibody (Amersham, Bioscience, Arlington Height, IL, USA) as the control, followed by treatment with 100 ng/mL of recombinant SerpinB3 protein to induce invasiveness [5]; (b) incubation with the LRP inhibitor RAP (5 or 50 g/mL, Meridian Life Science, Inc., Memphis, TN, USA) in the presence of 100 ng/mL of recombinant SerpinB3 protein; and (c) HepG2/control cells with medium and vehicle-only (PBS) as a negative control. The cell index of each well was measured every 10 min for up to 23 h at 37 °C in a 5% CO_2_ atmosphere using RTC software (version 2.0, ACEA BioSciences, San Diego, CA, USA).

### 2.4. SerpinB3 Quantification by ELISA

The protein concentration of cellular pellets was quantified by a bicinchoninic acid assay (BCA) protein kit (Pierce Merck, Darmstadt, Germany), following the manufacturer’s instructions and using BSA as a standard on a Victor X3 microplate reader (Perkin Elmer, Waltham, MA, USA).

SerpinB3 concentration was measured by a sandwich ELISA (HEPA Lisa kit, Xeptagen, VEGA Park, Venice, Italy), as previously described [18]. Briefly, undiluted samples were incubated for 1 h at room temperature on plates coated with rabbit anti-human SerpinB3 capture Ab (10 µg/mL in PBS, pH 7.4). A standard curve, obtained by the dilution of the recombinant SerpinB3 from 16 to 0.25 ng/mL, was also included. All samples were tested in duplicate. After washing, SerpinB3 was revealed by incubation with HRP-conjugated streptavidin secondary anti-SerpinB3 Ab (0.5 µg/mL). The plate was developed with a ready-to-use 3,30,5,50-tetramethylbenzidine (TMB) substrate solution. The reaction was stopped with 1 mol/L HCl, and absorbance at 450 nm was measured on a microplate reader (Victor X3; Perkin Elmer, Waltham, MA, USA).

### 2.5. Immunofluorescence

#### 2.5.1. Primary Human Monocytes and THP-1 Cells

Adherent primary human monocytes and THP-1 cells (3 × 10^5^ cells/slide) were seeded on slides and cultured for 24 h. In some experiments, the Wnt inhibitor ICG-001 (5 uM for 24 h, kindly provided by Dr. M. Cadamuro, University of Padua, Padua, Italy), was used to confirm the activation of the Wnt pathway [19]. Cells were fixed in 4% paraformaldehyde and permeabilized with 0,4% Triton X-100 and blocked with 5% goat serum (Invitrogen Life Technologies, Waltham, MA, USA) in PBS containing 1% BSA. Slides were incubated with the primary antibodies (SerpinB3, LRP1, LRP6, Wnt1, Wnt7a, β-catenin, axin, cMyc) for one hour at room temperature and then incubated for 1 h at room temperature with a mouse or a rabbit secondary antibody, based on the origin of the primary antibody. The cellular nuclei were counterstained by DAPI (Merck KGaA, Darmstadt, Germany), mounted with ELVANOL Merck KGaA, Darmstadt, Germany), and observed under a fluorescence microscope (Axiovert 200M-Apotome.2, Carl Zeiss MicroImaging GmbH, Göttingen, Germany). The characteristics of the primary and secondary antibodies used in the study are reported in Appendix A.

#### 2.5.2. HepG2 Cells

HepG2/control cells were seeded on slides (4 × 10^5^ cells/slide) and cultured for 24 h. Cells were treated as follows: (a) overnight incubation with PBS as a negative control, (b) overnight incubation with 100 ng/mL of SerpinB3 recombinant protein as a positive control, (c) pre-treatment with 5 g/mL of the anti-LRP-1 85 kDa monoclonal antibody or with a generic mouse Ig for one hour followed by overnight incubation with 100 ng/mL of SerpinB3 recombinant protein, and (d) overnight incubation with 5 g/mL of RAP and 100 ng/mL of SerpinB3 recombinant protein. Cells were then fixed with 4% paraformaldehyde, permeabilized with 0.4% Tryton X-100, and blocked with 5% goat serum (Invitrogen Life Technologies, Waltham, MA, USA) in PBS containing 1% BSA. Slides were incubated with polyclonal anti-vimentin and anti-snail antibodies and with monoclonal anti-E-cadherin antibodies for 1 h at room temperature, followed by incubation with the Alexa-Goat 546 and 488 secondary antibodies, respectively. Cellular nuclei were counterstained with Dapi (Merck KGaA, Darmstadt, Germany). Slides were mounted with ELVANOL (Merck KGaA, Darmstadt, Germany) and observed under a fluorescence microscope (Axiovert 200M-Apotome.2, Carl Zeiss MicroImaging GmbH, Göttingen, Germany).

### 2.6. ImageStream Analysis

To assess the ImageStream analysis, which combines high-resolution microscopy and flow cytometry, 4 × 10^6^ THP-1 cells, previously treated with SerpinB3 (100 ng/mL) in the presence of RAP (5 g/mL) or a medium alone for 24 h, were fixed and permeabilized with Fix and Perm (Invitrogen Life Technologies, Waltham, MA, USA), blocked with 5% goat serum (Invitrogen Life Technologies, Waltham, MA, USA) in PBS containing 1% BSA, and incubated with anti-LRP-1 for 1 h at room temperature. After washing, the cells were incubated for 30 min with secondary antibodies anti-mouse Alexa Fluor 488. Cellular nuclei were counterstained with Dapi by 3 min incubation (Merck KGaA, Darmstadt, Germany). For ImageStream analysis, 60 L of each sample was used, and the results were assessed by the IDEAS software (Luminex, Genk, Belgium).

### 2.7. RNA Isolation and Quantitative Real-Time PCR

Total RNA was extracted from cell lines and tissue samples using a Rnase Trizol reagent (Invitrogen, Carlsbad, CA, USA), according to the manufacturer’s instructions. After the determination of the purity and the integrity, total RNA, complementary DNA synthesis, and quantitative real-time PCR reactions were carried out, as previously described [6], using the CFX96 real-time instrument (Bio-Rad Laboratories Inc., Hercules, CA, USA). The housekeeping gene glyceraldehyde-3-phosphate dehydrogenase (GAPDH) was analyzed in all amplification sets to assess the integrity of total RNA. Primer sequences used in the study are reported in Appendix A.

### 2.8. Western Blot Analysis

The total protein content (50 µg) from each cellular extract, prepared at 4 °C in a RIPA lysis buffer in the presence of phosphatase and protease inhibitors (Roche, Indianapolis, IN, USA), was loaded onto a 10% polyacrylamide gel. The blots were probed with the following primary antibodies: anti-LRP-1 extracellular domain (85 kDa), anti-Vimentin, and anti-E-cadherin; mouse monoclonal anti-β-actin was used as the housekeeping control. Anti-mouse and anti-rabbit horseradish peroxidase-conjugated antibodies were used as secondary antibodies. Antigenic detection was carried out by enhanced chemiluminescence (Amersham, Arlington Heights, IL, USA), and densitometric analysis was assessed using the VersaDoc Imaging System (Bio-Rad Laboratories, Hercules, CA, USA). Relative density units were obtained by assessing the ratio of the intensity of the antigen and the housekeeping reference band (β-Actin). Moreover, a Western blotting analysis for β-catenin and SerpinB3 was performed on liver samples derived from an experimental murine model of hepatic carcinogenesis.

### 2.9. Mouse Model of Liver Carcinogenesis

In order to assess the in vivo effect of SerpinB3 on the Wnt/β-catenin axis, we performed analyses on liver cancer samples with different extents of SerpinB3 expression using fully characterized animal experimentation [20]. NAFLD-related liver carcinogenesis was induced in mice carrying a specific hepatocyte deletion of Hypoxia-inducible factor-2α (hHIF-2α-/- KO mice, n = 5) and in the related control of wild type littermates without HIF-2α deletion (WT, n = 7). These mice were subjected to an established experimental protocol involving a single administration of DEN (25 mg/kg body weight, intraperitoneally) at the age of 2 weeks, followed by feeding with a CDAA diet (Laboratorio Dottori Piccioni, Gessate, Italy) for 25 weeks starting from the age of 6 weeks [21]. Mice were kept under specific pathogen-free conditions and maintained with free access to pellet food and water. Liver samples were obtained and immediately frozen and thereafter maintained at −80 °C for further analysis. These experiments complied with EU and national ethical guidelines for animal experimentation, and experimental protocols were approved by the Animal Ethics Committee of the University of Oriental Piedmont, Novara, Italy, and the Italian Ministry of Health (authorization No. 1114/2016).

### 2.10. Immunohistochemistry

Immunostaining for Wnt-1 and the macrophage marker F4/80 was carried out as previously described [22]. Briefly, paraffin sections (2 μm thick), mounted on poly-L-lysine coated slides, were incubated with the following antibodies: a monoclonal antibody against Wnt-1 and a rabbit monoclonal antibody against F4/80. After blocking endogenous peroxidase activity with 3% hydrogen peroxide and performing microwave antigen retrieval, primary antibodies were labeled using EnVision, an HRP-labeled System (DAKO, Glostrup, Denmark), and visualized by a 3′-diaminobenzidine substrate. Conventional histological staining (Hematoxylin and Eosin, PAS stain for glycogen) was performed on paraffin sections (2 μm thick).

### 2.11. Patients

To confirm the effect of SerpinB3 on the Wnt/β-catenin axis, 38 surgically removed specimens of patients with hepatocellular carcinoma (HCC) were analyzed. Frozen liver tissue samples were maintained at −80 °C until use. The samples were obtained by the patients’ written consent from all subjects involved in the study, following a procedure that was approved by the Ethical Committee of the Padua Teaching Hospital (11 December 2006). The demographic and clinical profiles of the patients included in the study are depicted in Appendix A.

### 2.12. Amino Acid and Structure Alignment

The amino acid sequences of 19 human serpins were aligned using the Clustal Omega multiple sequence alignment.

The structured image was performed in PyMol 2.5.1 (Schrödinger). The SerpinB3 (PDB ID: 2ZV6) structure in its metastable conformation was retrieved from the Protein Data Bank.

### 2.13. Statistical Analysis

Statistical analysis was performed by Student’s *t*-test for analysis of variance when appropriate. Spearman’s rank correlation was used to measure the statistical dependence between two variables. All *p*-values reported were two-tailed and considered significant if *p* ≤ 0.05. Data in bar graphs are presented as mean ± SEM and were obtained from at least three independent experiments. Western blot and morphological images are representative of at least three experiments with similar results. Statistical tests were performed using GraphPad Prism and IDEAS software (Amnis-Imagestream Imaging Flow cytometer).

## 3. Results

### 3.1. Exogenous SerpinB3 Increases Monocyte Proliferation and Induces Its Endogenous Expression

Based on our previous studies on the increase in cellular proliferation promoted by SerpinB3 in hepatoma cells [15], we have evaluated whether this serpin is able to increase proliferation in monocytes, since monocytic cells are the main type of inflammatory cells in the tumor microenvironment and are profoundly involved in the pathogenesis and development of primary liver tumors, establishing a pro-inflammatory and pro-tumorigenic environment by the suppression of antitumor immune responses [23]. The results obtained in both primary human monocytes and in the THP-1 cell line demonstrate that the exogenous addition of SerpinB3 significantly increases the lifespan and the proliferation of both cell types. In particular, the addition of recombinant SerpinB3 at 100 ng/mL resulted in a significant increase in the replication of primary monocytes, starting on the fifth day of culture (Figure 1A). The positive effect of this serpin was confirmed in THP-1 cells, in which increased proliferation was achieved starting on the second day of culture (Figure 1B). It is worth noting that increased monocyte proliferation promoted by exogenous SerpinB3 was associated with an increase in its endogenous expression, as documented by IF (Figure 1C) and ELISA assays (Figure 1D), leading to a paracrine positive loop induction.

### 3.2. SerpinB3 Induces Wnt-1 and Wnt-7a

Based on the fact that SerpinB3 is able to increase β-catenin expression [11], we have evaluated whether this phenomenon can occur through the induction of members of the Wnt family, which are molecules responsible for activating catenin signaling. Primary human monocytes were stimulated with recombinant SerpinB3, and cells were harvested after 2 and 8 h to evaluate the mRNA expression of Wnt-1, Wnt-3a, Wnt-5a, Wnt-5b, and Wnt-7a genes. Interestingly, the cells treated with SerpinB3 showed a significant upregulation of Wnt-1, already detectable at 2 h (Figure 2A), and Wnt-7a, which was detectable at 8 h (Figure 2B). By contrast, no increase was observed for the expression of Wnt-3a, Wnt5a, and Wnt-5b (data not shown). An immunofluorescence analysis of Wnt-1 confirmed the ability of SerpinB3 to induce the expression of this protein that remained remarkably high for up to 72 h (Figure 2C).

### 3.3. SerpinB3 Activates the Wnt-Canonical Pathway

Since Wnt-1 and Wnt-7a are the key players in the activation of the Wnt-canonical pathway [24,25], the expression of β-catenin was evaluated by immunofluorescence staining in monocytes treated or not with SerpinB3. After 24 h of incubation with this serpin, a remarkable increase in β-catenin at the cytoplasmic level was observed only in cells treated with SerpinB3, while axin, the negative regulator of the Wnt signaling pathway, was reduced, confirming the positive modulation of this pathway by SerpinB3 (Figure 3A). Accordingly, SerpinB3-induced monocyte proliferation was abrogated by the Wnt inhibitor ICG-001 (Figure 3B).

To confirm the ability of SerpinB3 to activate the Wnt pathway and to induce monocyte proliferation, we evaluated c-Myc expression, a downstream gene of the same pathway. As observed in Figure 4A, treatment of THP-1 cells with human recombinant SerpinB3 led to the upregulation of c-Myc mRNA expression in a cyclic manner over time. Accordingly, the specific Wnt inhibitor ICG-001 was able to abrogate the upregulation of c-Myc protein expression induced by SerpinB3 in primary human monocytes (Figure 4B).

### 3.4. SerpinB3 and Wnt/β-Catenin Axis in an Experimental Model of Liver Carcinogenesis

In order to demonstrate the relation between SerpinB3 and the Wnt/β-catenin pathway at the mechanistic level in liver carcinogenesis, we used liver tumor samples with different extents of SerpinB3 expression, derived from mice carrying a specific hepatocyte deletion of HIF-2α (hHIF-2α-/- KO mice), a previously recognized transcriptor factor that upregulates SerpinB3 by binding to its promoter [26].

As shown in Figure 5A, HIF-2α-/- KO mice, which in the previous study presented a significant decrease in the volume and number of liver tumors compared with the controls [20], showed a remarkable reduction in SerpinB3 and of β-catenin protein levels, compared to wild type mice. Of relevance, β-catenin protein levels were significantly higher in mice expressing high levels of SB3 (Figure 5B) and correlated with SerpinB3 (Figure 5C), confirming a positive relation between SerpinB3 and the Wnt/β-catenin pathway.

### 3.5. SerpinB3 and Wnt Family Members in Human Liver Tumors

Activation of the Wnt family members in relation to SerpinB3 expression was also assessed in the tumor tissue of 38 patients with HCC. When samples were divided according to the extent of SerpinB3 expression, tumors expressing a high level of SerpinB3 (>median value) showed a trend toward higher amounts of Wnt-1 and Wnt-7a compared to the corresponding figures observed in tumors with low SB3 level (<median value), although the difference did not reach statistical significance, probably due to the small sample size (Appendix A). Again, no differences were observed for Wnt-5a and Wnt-3a mRNA between the two groups (data not shown). These data suggest that SerpinB3 is able to upregulate Wnt signaling not only in vitro but also in a more complex environment, such as liver cancer.

### 3.6. SerpinB3 Upregulates Low-Density Lipoprotein Receptors

Since members of low-density lipoprotein receptors (LRPs), particularly LRP-5/6, are key co-receptors for the activation of canonical Wnt signaling, we have evaluated the effect of SerpinB3 on LRP-6 expression in primary human monocytes. An immunofluorescence analysis showed a significant increase in LRP-6 in cells treated with SerpinB3, compared to untreated cells, for up to 7 days (Figure 6A). These results were also confirmed in the THP-1 cell line exposed to SerpinB3, where the upregulation of LRP-6 was inhibited by treatment with the receptor-associated protein (RAP), a known inhibitor of LRP family members [27].

Since LRP-1 has been involved in cellular internalization and subsequent degradation of serine proteinases [13], we have explored whether this member of the LRP family was upregulated by SerpinB3. In THP-1 cells, SerpinB3 induced an upregulation of LRP-1 similar to that observed for LRP-6, which also in this case was inhibited by RAP, both at transcription and protein levels (Figure 6C,D).

The effect of SerpinB3 on the expression of the LRP family members was also confirmed in HepG2 cells, where cells transfected to overexpress SerpinB3 showed upregulation of LRP family members, including LRP-1, LRP-5, and LRP-6 (Figure 7 and Appendix A). In HCC specimens, in addition to the described trend toward a higher activation of the Wnt signaling in relation to SerpinB3 expression, statistically significant higher levels of LRP-1 transcription were found in tumors expressing high levels of SerpinB3 (Appendix A).

### 3.7. Amino Acid and Structure Alignment of Serpins

In order to better assess the interaction of SerpinB3 with LRP-1, which is known to bind serine-proteinase complexes, we aligned the amino acid sequences of SerpinB3, alpha-1 antitrypsin (AAT), and several other human serpins in the region comprising the SEC binding pentapeptide described by Joslin et al. [28]. We found that the SerpinB3 369-FLFFI–373 pentapeptide, even if different from the one originally described in AAT, is highly conserved in Serpins, in particular in the SerpinB family, in nexin 1 (UniProt ID P07093, serpin E2) and in Alpha-2-antiplasmin (UniProt ID P08697, serpin F2) (Figure 8A). In relation to the aa sequence of SerpinB3, this region is located closely downstream of the reactive site loop (Figure 8B).

Along the aligned human Serpins, F369 is fully conserved, and residues 370–373 correspond to amino acid groups of similar properties (i.e., hydrophobic). It is worth noting that structure alignment between SerpinB3 (PDB ID: 2ZV6) in the 336–379 region and the putative LRP-1 binding site of the AAT (29) (PDB ID: 1QLP) 364–380 region (peptides FHCNHPFLFFIRQNKTN and VKFNKPFVFLMIEQNTK, respectively) shows an identical secondary structure organization, indicating a similar backbone organization in the selected regions of the two proteins (Figure 9A). As for the pentapeptide region, an overlap between the side chains of the first, highly conserved, and third phenylalanine can be observed, while SerpinB3 position 2 is occupied by an L instead of a V, and position 4 is occupied by a third F in place of an L, amino acids with similar properties in both cases (Figure 9B). Taken together, these observations suggest the possibility of a tridimensional organization in SerpinB3 pentapeptide compatible with the one described for AAT and relevant for binding to LPR-1, which is predicted to occur in the same binding site of LRP-1 by computational modeling (Figure 9B).

Moreover, in position 5 of the pentapeptide, SerpinB3 has a naturally occurring I, instead of M found in AAT, a replacement known to improve its stability and binding capability to LPR-1 [29]. Therefore, it is possible to speculate that the natural binding of SerpinB3 to LPR-1 would be favored if compared to the one described for AAT.

### 3.8. SerpinB3 Induces the Pro-Invasive Activity of LRP-1

A growing amount of evidence has strengthened the putative role of LRP-1 in crucial events during cancer progression by promoting cell migration and invasion [30], in addition to cell survival [14]. In order to evaluate whether SerpinB3 could induce these biological activities in hepatocytes, we have used stably transfected HepG2 cells overexpressing SerpinB3, which showed a remarkable increase in LRP-1 at the transcription and protein levels (Figure 7 and Appendix A). It is worth noting that the already reported invasion capability induced by SerpinB3 in hepatocytes [5] and mouse fibroblasts [17] was significantly reduced in HepG2 cells treated with SerpinB3 by both an anti-LRP-1 antibody and RAP in a dose–dependent manner (Figure 10A). These results were associated with a parallel reduction not only of β-catenin but also of SNAIL and vimentin, together with the increased expression of E-cadherin (Figure 10B), as confirmed by a Western blot analysis (Appendix A), supporting the hypothesis that the EMT profile induced by SerpinB3 can be reverted by the ligand inhibition of LRP family members.

## 4. Discussion

Wnt-β-catenin signaling is known to regulate multiple cellular processes, including embryonic development and adult tissue homeostasis. However, the abnormal regulation of this signaling pathway is also involved in the development of several chronic diseases and tumors of different organs. Indeed, once activated, the Wnt pathway induces the increased stability of β-catenin and its nuclear translocation, which facilitates the expression of genes involved in cell proliferation, survival, differentiation, and migration [24].

In the present study, we have evaluated the ability of SerpinB3 to modulate the Wnt pathway in in vitro and in vivo models. In primary human monocytes and the monocytic cell line THP-1, the addition of SerpinB3 not only significantly increased cell lifespan and proliferation, but also resulted in a positive modulation of the Wnt pathway. SerpinB3 induced a significant upregulation of Wnt-1/7 and of β-catenin, mainly localized in the nucleus, confirming its activation profile. These findings were also supported by the results obtained in experimental carcinogenesis, where liver tumors of hHIF-2 -/- KO mice presented not only a remarkable reduction in SerpinB3 expression, as a result of the lack of HIF-2a transcriptional activity for SerpinB3 [26], but also a parallel decrease in β-catenin expression, confirming the role of this serpin in the activation of Wnt pathway.

To further understand the mechanism of Wnt activation by SerpinB3, we focused our attention on LRP family members, initially on LRP-5 and LRP-6, which have been described as cellular receptors for Wnt pathway activation [25]. The results obtained have demonstrated that both receptors were upregulated by SerpinB3. We further expanded our study to another member of the LRPs family, namely LRP-1, since it was previously described that this receptor binds several protease complexed-serine protease inhibitors [13]. According to the hypothesis, this receptor was found to be upregulated by SerpinB3. In addition, computational analysis and 3D modeling revealed that the putative SerpinB3 binding site could determine an even more stable interaction with LRP-1 than initially described as responsible for the binding of the enzyme-complexed serpins with the cellular receptor [28], which was further expanded by Toldo et al. for AAT [29]. This hypothesis is supported by the fact that an AAT peptide, including a similar and structurally compatible aa sequence, is able to bind and activate LRP-1-mediated signaling [29]. It is interesting to note that the AAT pentapeptide neo-domain was identical in SerpinB3 and in Nexin-1, where it was involved in the invasion activity and metastasis formation induced by this serpin [31]. In addition, one should note that the LRP binding site of SerpinB3 is located downstream of the reactive site loop of the protein, and this fact supports our previous findings reporting that loop-deleted SerpinB3 had effects similar to those of wild type SerpinB3 on the expression of EMT markers and β-catenin expression [5]. The inhibition of LRP-1 by the corresponding antibody or by the use of the LRP pan-inhibitor RAP was able not only to determine a decrease in LRP cell surface expression but also a dose–dependent reduction in SerpinB3-induced invasiveness. These results were in line with the observed reversal of the expression of the SerpinB3-induced EMT profile markers by both the anti-LRP-1 antibody and RAP.

## 5. Conclusions

SerpinB3 upregulates LRP family members and determines not only the activation of the Wnt canonical pathway but also the activation of LRP-1, a cellular receptor profoundly implicated in cell survival and invasiveness. These findings could provide useful information for the development of targeted therapeutic strategies in an oncological setting.

## Figures and Tables

**Figure 1 biology-12-00771-f001:**
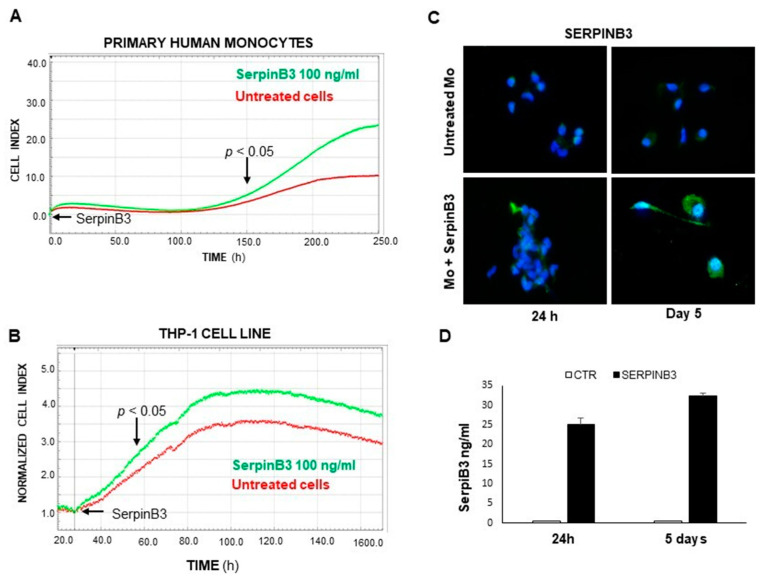
The Effect of SerpinB3 on the proliferation of monocytic cells. An example of a direct comparison of real-time growth curves, generated by the xCELLigence RTCA of primary human monocytes (**A**) and THP-1 cell lines (**B**) stimulated with 100 ng/mL of recombinant SerpinB3 vs. untreated cells. (**C**) An immunofluorescence analysis of the expression of SerpinB3 in primary human monocytes after 24 h and 5 days of treatment. Blue color: DAPI nuclear staining; green color: SerpinB3 staining. Original magnification 63×. (**D**) A time course quantification of SerpinB3 protein by ELISA in cellular extracts of primary monocytes after 24 h and 5 days of treatment. All graphs represent mean results from three independent experiments ± SD.

**Figure 2 biology-12-00771-f002:**
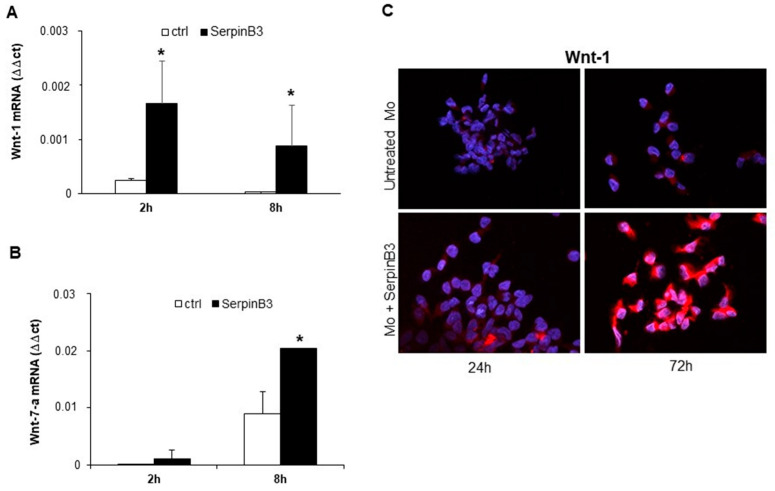
Wnt-1 and Wnt-7-a expression. Relative mRNA expression of Wnt-1 (**A**) and Wnt-7-a (**B**) in primary human monocytes (Mo) stimulated with 100 ng/mL of SerpinB3, calculated by the Ct method. Data were normalized to housekeeping gene glyceraldehyde-3-phosphate dehydrogenase (GAPDH). All graphs represent mean results from three independent experiments ± SD. * *p* < 0.05 vs. the control. (**C**) An immunofluorescence staining of Wnt-1 in primary human monocytes after 24 and 72 h of treatment with 100 ng/mL of human recombinant SerpinB3. Blue color: DAPI nuclear staining; red color: Wnt-1 staining. Original magnification 63×. * *p* < 0.05 vs. control.

**Figure 3 biology-12-00771-f003:**
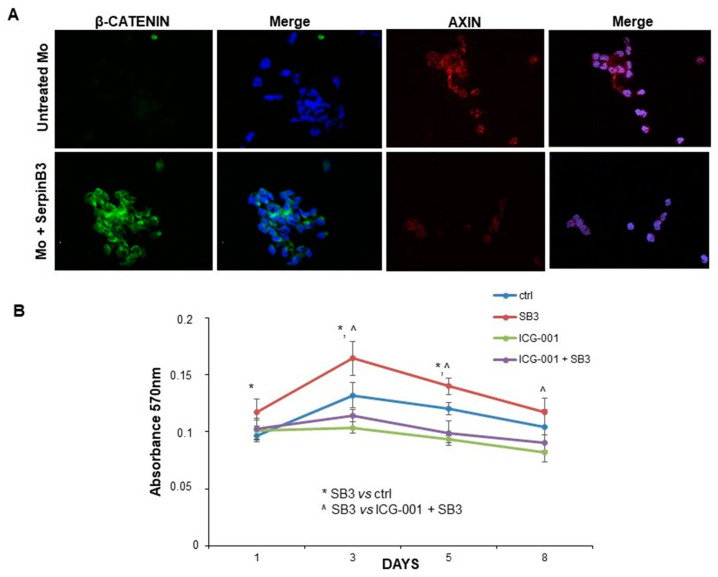
Expression of β-catenin and axin. (**A**) An immunofluorescence analysis of the expression of β-catenin and axin in monocytes treated with human recombinant SerpinB3. Cellular nuclei were counterstained with DAPI (blue). Original magnification 63×. (**B**) A time course analysis of the proliferation of monocytes treated with SerpinB3 +/−, the Wnt inhibitor ICG-001 by the MTT assay. * *p* < 0.05 vs. control, ^ *p* < 0.05 vs. ICG-001 + SB3.

**Figure 4 biology-12-00771-f004:**
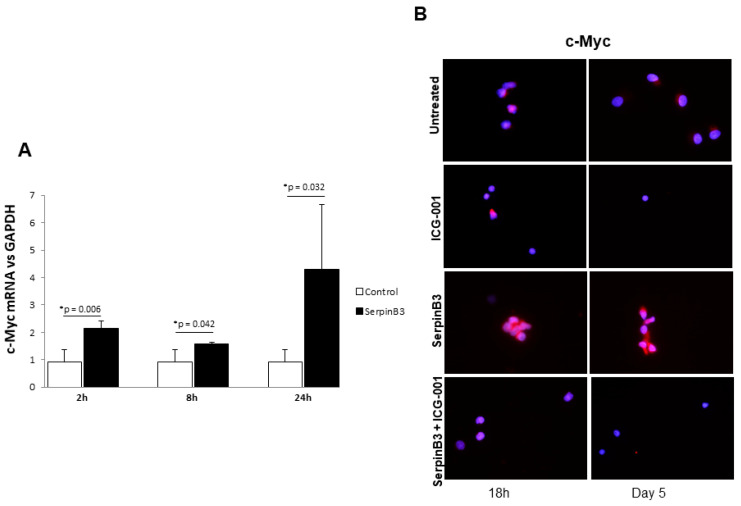
c-Myc expression in monocytic cells. (**A**) A time course analysis of c-Myc expression in THP-1 cells treated for 2, 8, and 24 h with 100 ng/mL of human recombinant SerpinB3. All graphs represent mean results from three independent experiments ± SD. *p*-values are indicated. (**B**) An immunofluorescence analysis of c-Myc +/− SerpinB3 or ICG-001 or SerpinB3 + ICG-001 in primary human monocytes after 18 h and 5 days of treatment. Blue color: DAPI nuclear staining; red color: c-Myc staining. Original magnification 63×.

**Figure 5 biology-12-00771-f005:**
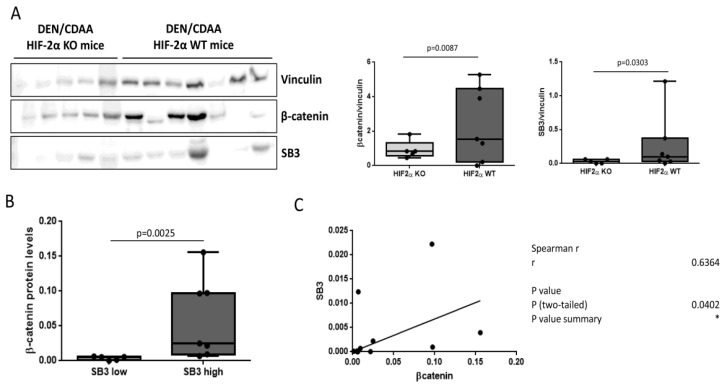
Correlation between SerpinB3 and β-catenin in DEN–CDAA liver carcinogenesis. (**A**) A Western blot analysis of SerpinB3 (SB3) and β-catenin protein levels in tumor masses obtained from HIF-2α wild type (WT, n = 7) or HIF-2α-/- KO (n = 5) mice treated with the DEN/CDAA protocol. For the Western blot analysis, Bio-Rad (Hercules, CA, USA) Quantity One software was used to perform the densitometric analysis. Equal loading was evaluated by reprobing membranes for Vinculin. Statistical differences were assessed by the Mann–Whitney test for nonparametric values. (**B**,**C**) The relationship between SerpinB3 and β-catenin protein levels in liver tumors, including all mice. Higher β-catenin protein levels were obtained in mice with high levels (>median values) of SerpinB3, compared to those with low (≤median values) SerpinB3 values. A correlation analysis between SerpinB3 and β-catenin was performed with the Spearman r test. In panel (**C**) *p* value is specified (*p* = 0.0402) and * indicates that it is < 0.05.

**Figure 6 biology-12-00771-f006:**
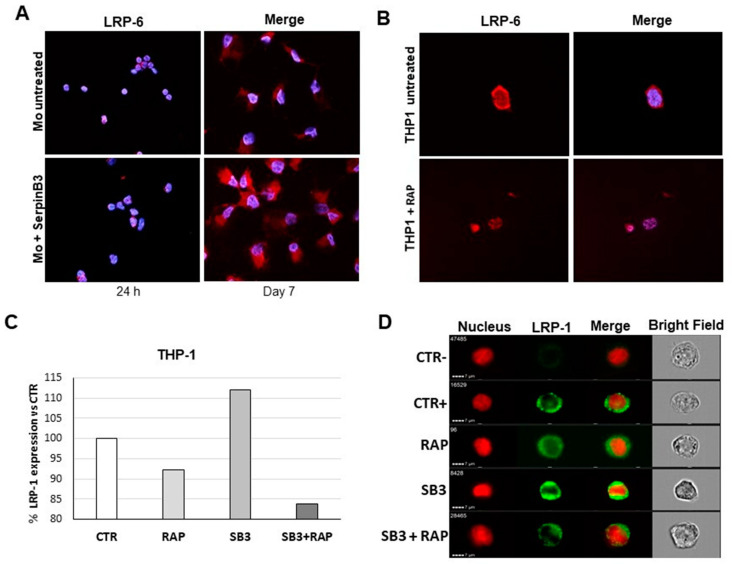
LRP-6 and LRP-1 expression in monocytic cells. (**A**) An immunofluorescence analysis of LRP-6 in human primary monocytes stimulated with 100 ng/mL of recombinant SerpinB3 for 24 h and 7 days. Blue color: DAPI nuclear staining; red color: LRP-6 staining. (**B**) An immunofluorescence analysis of LRP-6 in THP-1 cells, physiologically expressing SerpinB3, +/− RAP. Blue color: DAPI nuclear staining; red color: LRP-6 staining. Original magnification 63×. (**C**) The percentage of LRP-1 expression in THP-1 cells +/− SerpinB3 or SerpinB3 + RAP or RAP alone. Results are expressed as a percentage of LRP1 expression compared to the untreated cells. (**D**) Representative images of cellular expression of LRP1 in THP-1 cells +/− overnight incubation with SerpinB3 (100 ng/mL) or SerpinB3 + RAP or RAP (5 ug/mL) alone, generated by an ImageStream Flow Cytometer.

**Figure 7 biology-12-00771-f007:**
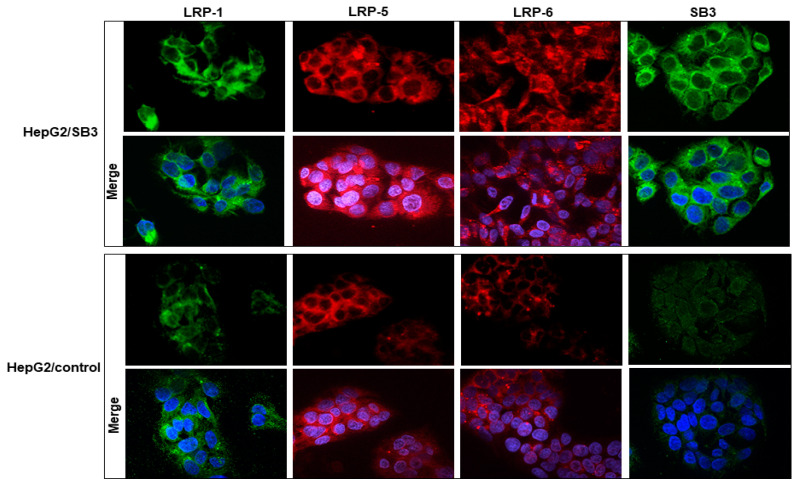
LRP expression in hepatoma cells. An example of an immunofluorescence analysis of LRP-1 (green), LRP-5 (red), LRP-6 (red), and SerpinB3 (green) in HepG2 overexpressing SerpinB3 (HepG2/SB3) compared with the HepG2/control cells tested 48 h from seeding. Cellular nuclei were counterstained with DAPI (blue). Original magnification 63×.

**Figure 8 biology-12-00771-f008:**
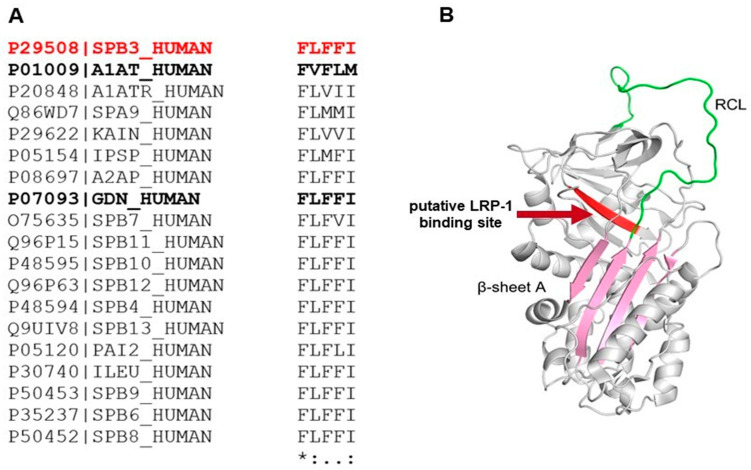
Alignment of the human serpin pentapeptide region and the structure of SerpinB3. (**A**) Amino acid sequence alignments of the human serpin pentapeptide region. In red, SerpinB3 sequence; in bold, α1-Antitrypsin (A1AT) and protease nexin-1 (GDN) sequences. * Positions with a single, fully conserved residue. : Positions with a conservation between amino acid groups of similar properties. (period) Positions with a conservation between amino acid groups of weakly similar properties. (**B**) The structure of native SerpinB3 (PDB ID: 2ZV6) represented as a cartoon. The β-strand containing the LRP-1 binding pentapeptide is in red. The reactive center loop (RCL) is in green. The functionally relevant β-sheet A is in pink.

**Figure 9 biology-12-00771-f009:**
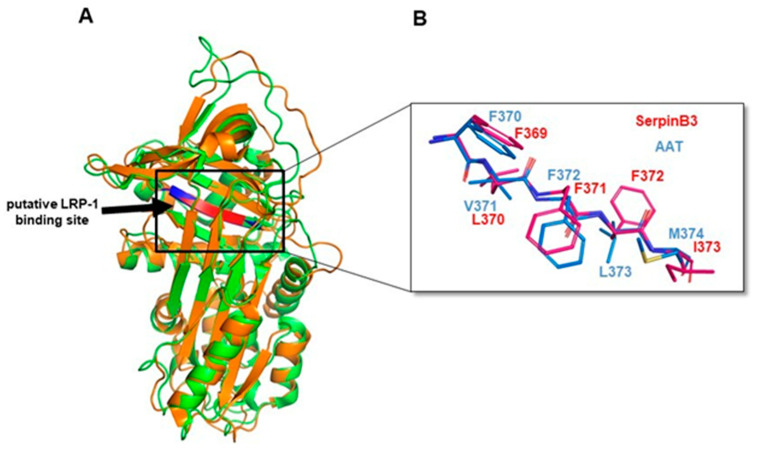
A Cartoon of SerpinB3-LRP-1 binding. (**A**) The structural alignment of native SerpinB3 (PDB ID: 2ZV6, green) and AAT (PDB ID: 1QLP, orange) represented as a cartoon. The β-strand containing the putative LRP-1 binding region is colored in red in SerpinB3 and blue in AAT. (**B**) Magnification of the structure alignment of native SerpinB3 (PDB ID: 2ZV6, red) and AAT (PDB ID: 1QLP, blue) in the putative LRP-1 binding pentapeptide region represented as sticks. Amino acids are labeled according to primary sequence numbering.

**Figure 10 biology-12-00771-f010:**
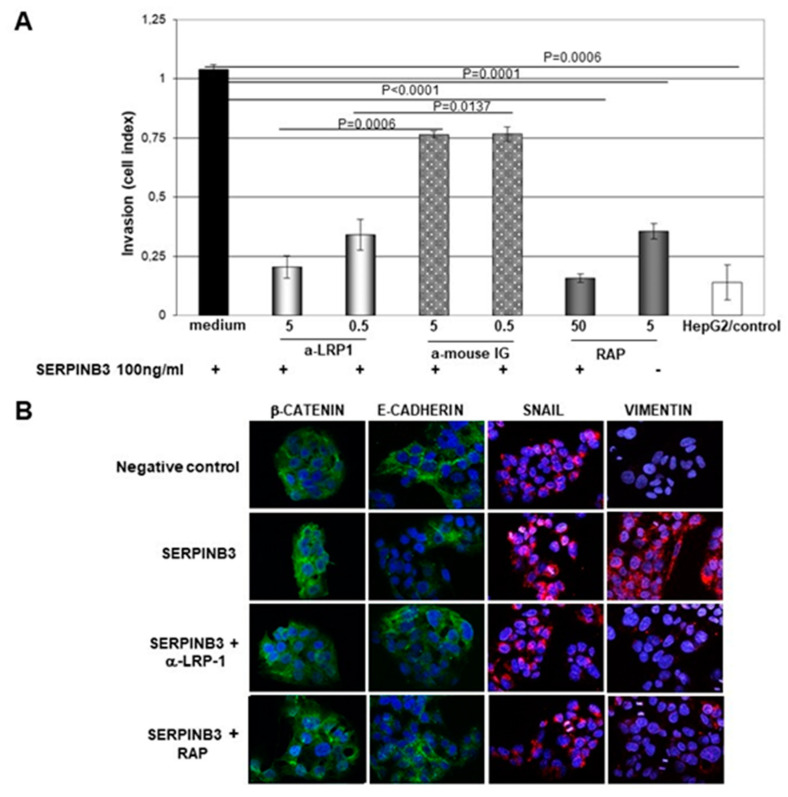
Real-time monitoring of cell invasion. (**A**) An invasion analysis of HepG2 cells treated with human recombinant SerpinB3 (100 ng/mL) with or without one-hour pre-treatment with 0.5 or 5 g/mL of anti-LRP-1, with 5 or 50 g/mL of RAP and PBS as the control. The invasion curves were monitored every 10 min for 23 h, using xCELLigence RTCA (ACEA, San Diego, CA, USA). Experiments were performed in quadruplicate. Invasion bars are depicted as mean ± SD cell index values of four wells/treatment processed in parallel at 10 h time points. (**B**) An immunofluorescence analysis of β-catenin (green), E-cadherin (green), snail (red), and vimentin (red) in HepG2/empty vector cells incubated overnight with PBS, as a negative control, with 100 ng/mL of recombinant SERPINB3 protein, as a positive control, pre-treated with 5 g/mL of anti-LRP-1 for 1 h, incubated overnight with 5 g/mL of RAP or with 100 ng/mL of recombinant SerpinB3. Blue color: DAPI nuclear staining. Original magnification 63×.

## Data Availability

The data presented in this study are available upon request from the corresponding author. The data are not publicly available due to privacy reasons.

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
