# Peer review of "SerpinB3 Upregulates Low-Density Lipoprotein Receptor-Related Protein (LRP) Family Members, Leading to Wnt Signaling Activation and Increased Cell Survival and Invasiveness"

_biology, 2023, doi:10.3390/biology12060771_

Round 1

Reviewer 1 Report

In this manuscript, the authors investigated the ability of SerpinB3 to modulate the Wnt pathway in liver cancer. In a first part, they demonstrated that SerpinB3 was able to activate the canonical Wnt pathway in primary human monocytes, by increasing the expression of WNT1 and WNT7, as well as the level of beta-catenin protein in the cytoplasm and the expression of c-MYC, one of its target genes. To go further, they studied the effect of serpinB3 on the expression of low-density lipoprotein receptors (LRPs) that are co-receptors for the activation of the canonical Wnt pathway. They found that LRP6 and LRP1 exhibited an increased expression at protein level in monocytic cells treated with recombinant serpinB3.

Then, they showed a correlation between the expression of serpinB3 and beta-catenin, at the protein level, in a mouse model of liver carcinogenesis. By using the HepG2 hepatoma cell line transfected with a serpinB3 expression vector, they confirmed the results obtained in monocytic cells, ie the expression of serpinB3 resulted in the overexpression of LRP1 and LRP6. As LRP-1 is able to bind serine-protease complexes, the authors looked for a possible interaction between serpinB3 and LRP1. By comparing the aminoacid sequence and secondary structure of serpinB3 and α1-Antitrypsin (AAT), they concluded that the serpinB3 tridimensional organization was compatible with its possible interaction with LRP1. Because LRP1 has been reported to be involved in cell survival, but also in cell migration and invasion, they looked for the invasion capacity of cells. They showed that the invasion capacity of serpinB3-treated HepG2 cells was dramatically enhanced compared with HepG2 cells, and reduced in presence of either an anti-LRP1 antibody or a LRP inhibitor.

Altogether, their results support a role of serpinB3 in liver carcinogenesis via the activation of the Wnt signaling pathway, in particular the upregulation of LRP1, implicated in cell survival and invasiveness.

Major modifications:

ü    While describing interesting results, the manuscript gives the impression of an unfinished process and lacks a link:

1-      In the simple summary, the authors wrote: “the aim of our work was to evaluate the ability of SerpinB3 to modulate the Wnt pathway in liver cancer”. But, five out of ten figures show results obtained in monocytic cells, without a clear link being made, in the Results part of the manuscript, between these cells and the liver cancer cells.

2-      By using a cohort of 38 HCC samples, they showed that the expression of Wnt1 and Wnt7 is higher in HCC with a high expression of serpinB3, but the correlation is not statistically significant, probably due to the small sample size. Why did not they extend their study by using the TCGA-LIHC database?

3-      From an in silico analysis, the authors suggest that serpinB3 is able to interact with LRP. To confirm this, they could have done a proximity ligation assay.

4-      In the last part of the manuscript, the author focus on the upregulation of LRP1 by serpinB3 and on the consequences on the invasion capacity of HepG2 cells. All the experiments were performed in only one liver cancer cell line. The authors could have looked at whether LRP1 is found more expressed in HCC samples with a high expression of serpinB3 (in their own HCC cohort and the TCGA-LIHC database).

ü    To investigate the modulation of the canonical Wnt pathway, the authors use a hepatoma cell line, in which the pathway is already activated, because this cell line expresses a stabilized form of beta-catenin. The use of an additional cell line is essential to confirm the results from HepG2 cells.

ü    How to explain the dramatic decrease of several proteins (ex. Myc, Fig.4) after 8h of serpin treatment on non-synchronized cells, since the effect on proliferation is visible until 5 days?

Minor modifications

- Fig. 1: “D) time course quantification of SerpinB3 protein by ELISA in primary monocytes supernatant after 24 hours and 5 days of treatment”. How to distinguish endogenous and exogenous SerpinB3? What is the half-life of SerpinB3 in the culture medium?

- Fig.7: Why use a transfected serpinB3 vector and not the recombinant SerpinB3?

- End of p4: “with 0.4% Tryton X-100”. Please correct

- Please check for “b” in the main text and supplementary material. It appears sometimes in a wrong form. Ex: p5, 2.9. Mouse model of liver carginogenesis: “Wnt/-catenin axis”

- Histograms: Fig.4: The Y axis represents a ratio, not the DDCT. Please correct

- Immunofluorescence/ Please indicate the magnification for figures 2 and 3.

- Table S3: GII grade is noted two times

- End of the introduction: “an-titrypsin”. Please correct

- “A growing number of evidences have strengthened the putative role of LRP-1 in crucial events during cancer progression by promoting cell migration and invasion [26], beside cell survival [14]”. Other more recent references could be useful, in particular in relation with EMT = Chiu et al., doi: 10.3390/cancers12040847; Tian et al., doi: 10.1242/jcs.228213.

Reviewer 2 Report

Reviewer Comments

Journal: Biology

Paper number: biology-2298311

Paper title: SerpinB3 upregulates low-density lipoprotein receptor-related protein (LRP) family members leading to Wnt signaling activation and increased cell survival and invasiveness.

Authors: Santina Quarta, Andrea Cappon, Cristian Turato, Mariagrazia Ruvoletto, Stefania Cannito, Gianmarco Villano, Alessandra Biasiolo, Maristella Maggi, Francesca Protopapa, Loris Bertazza, Silvano Fasolato, Maurizio Parola, Patrizia Pontisso 

General Comment:

The research paper under revision is seeking to understand the interaction of the protease inhibitor SerpinB3 with low-density lipoprotein receptor-related protein (LRP) family members involved in liver cancer. In particular, the Authors focused on the upregulation of co-receptors LRP-5/6 implicated in cell survival and invasiveness which have been described as cellular receptors for aberrant Wnt/β-catenin signaling pathway activation. Then, better understanding of SerpinB3 as agonist and/or activator of aberrant Wnt/β-catenin pathway could rationalize innovative strategies for cancer treatment.

There are numerous strengths to this study, including in vitro and in vivo data and well-updated references. However, it requires some additional revision.

My minor criticisms relate to the following issues:

1.              Figure legends of Figs. 8, 9 require adjustment with the fonts. Currently fonts of figure legends 1, 2, 3, 4, 5, 6, 7 and 10 are different from Figs. 8 and 9.

2.              Pictures in panels Fig.1C, 2C, 3A, 4A, 6A, B, 10B are poor quality and need to be improved;

3.              In figure legends 2C, 3A add original magnification;

4.              In Figs. 7 cellular nuclei are not counterstained by DAPI. Please add DAPI staining;

5.              In Fig. 5A right panel fonts are too small and hard to read them. Adjustments are required. 

Reviewer 3 Report

The authors of this paper evaluated the ability of SerpinB3 to modulate the Wnt pathway in several in vivo and in vitro models. They propose that SerpinB3 modulates the Wnt signaling pathway by upregulating the Wnt co-receptors LRP-5, LRP-6, and LRP-1, which in turn promote cell survival and invasiveness. This study was well designed. However, in several places, the quality of data was not satisfactory and must be improved for publication.

1. In Figure 3, it is not very clear if the increased b-catenin staining is in the nucleus or cytoplasm. Separating the b-catenin and nuclear staining channels may help solve this issue. Adding an inset with zoomed in pictures may also help.

2. The quality of the entire Figure.5 was not satisfactory. In panel A, the loading control shows too much variability. In panel B, only the data points that close to 0,0 fit the regression model, 4-5 data points are far away from the model. The p value of 0.04 also suggests a poor fitting.  

3. The results of SerpinB3 and Wnt family members in human liver tumors did not show statistically significant difference, therefore, the authors should not conclude that Wnt-1 and Wnt-7a are higher in tumors.

4. The increase of LRP-6 and 5 were only shown with immunofluorescence, which is prone to batch-to-batch variation and therefore not sufficient to show quantitative difference. Western blots should be used to show increase more definitively.

5. Increase of LRP-1 was shown with western blots in S2 in HepG2 cells. However, the bands for LRP-1 was very diffusive and have very high backgrounds.

Round 2

Reviewer 1 Report

The modifications made in the manuscript and the authors’ response are generally satisfactory. Nevertheless, one point requires attention and another one improvement:

- In their response to comment 1, the authors write that they found an increase of SerpinB3 in only 4 tumors out of the 369 included in the TCGA-LIHC database. How do they explain this result? It does not seem that the characteristics of their cohort are very different from those of the TCGA. Could expression detection methods be the cause? Or the presence of many monocytic cells in the tumors. It might be interesting to look at the anatomical pathology reports and if so, to consider them in the manuscript.

- The authors ‘response to comment 2 is not satisfactory. Indeed, both c-Myc mRNA and protein have a half-life around 30 minutes. Consequently, only a time course analysis that would show that the induction of c-Myc mRNA is cyclic could allow understanding the difference in mRNA and protein expression shown on Figure 4. This time course should be added.

Reviewer 3 Report

The authors have addressed the majority of the concerns.

Author Response

We are grateful to the reviewer for his overall consideration of the revised paper. As for his suggestion to check minor spelling, we have carefully reviewed the manuscript, accordingly.